# Challenges and Future Perspectives of Promising Biotechnologies for Lignocellulosic Biorefinery

**DOI:** 10.3390/molecules26175411

**Published:** 2021-09-06

**Authors:** Yansong Liu, Yunhan Tang, Haiyan Gao, Wenming Zhang, Yujia Jiang, Fengxue Xin, Min Jiang

**Affiliations:** 1State Key Laboratory of Materials-Oriented Chemical Engineering, College of Biotechnology and Pharmaceutical Engineering, Nanjing Tech University, Nanjing 211800, China; 13852346146@163.com (Y.L.); tyh_491510410@163.com (Y.T.); gaohaiyan06@163.com (H.G.); zhangwm@njtech.edu.cn (W.Z.); bioengine@njtech.edu.cn (M.J.); 2Jiangsu National Synergetic Innovation Center for Advanced Materials (SICAM), Nanjing Tech University, Nanjing 211800, China

**Keywords:** lignocellulose, biorefinery, enzymatic hydrolysis, microbial fermentation

## Abstract

Lignocellulose is a kind of renewable bioresource containing abundant polysaccharides, which can be used for biochemicals and biofuels production. However, the complex structure hinders the final efficiency of lignocellulosic biorefinery. This review comprehensively summarizes the hydrolases and typical microorganisms for lignocellulosic degradation. Moreover, the commonly used bioprocesses for lignocellulosic biorefinery are also discussed, including separated hydrolysis and fermentation, simultaneous saccharification and fermentation and consolidated bioprocessing. Among these methods, construction of microbial co-culturing systems via consolidated bioprocessing is regarded as a potential strategy to efficiently produce biochemicals and biofuels, providing theoretical direction for constructing efficient and stable biorefinery process system in the future.

## 1. Introduction

It is known that more than 90% of hydrocarbons can be refined from fossil resources; however, the wide usage of fossil resources results in the price fluctuation, serious environmental pollution and energy crises [1]. Biorefinery is considered as the potential process to replace petroleum refinery, in which biofuels and biochemicals can be produced from renewable bioresources [2]. Sugar-based agricultural materials are commonly used as the feedstocks for biofuels production in first-generation biorefinery, including starch, sugarcane or rapeseed, which can be easily extracted by squeezing, water-based extraction, adding amylase, etc. Currently, there are almost 370 plants focusing on first-generation biorefinery in Brazil with a total capacity of 43,105 million liters of ethanol from sugarcane [3]. In the EU, biodiesel production achieved about 11 million tonnes from various vegetable oils [4]. Although the technology of first-generation biorefinery has significantly developed for biodiesel and bioethanol production, the large demands for agricultural materials will lead to unstable production costs. Thus, exploration of non-edible agricultural wastes as feedstock has attracted more and more attention [5].

Lignocellulose is a kind of promising resource for biorefinery owing to their abundance, renewability and non-competition with human demands. Their output can reach 170 billion tons annually. However, only 3% of lignocellulose can be effectively utilized in the circular bioeconomy [6]. Hence, rational and efficient utilization of lignocellulose has always been a hot research topic. Lignocellulose derived from wastes or residues of agricultural and forestry industry activities is mainly composed of cellulose (35–50%), hemicellulose (23–32%) and lignin (15–30%) [7,8]. These cellulosic polysaccharides and non-cellulosic polysaccharides constitute 90% of plant dry weight (Figure 1). Cellulose is composed of thousands of *D*-glucose bound by *β*-1,4 linkages [9]. The multiple linear cellulose molecules are closely bound to form a highly crystallized microfiber structure by intermolecular hydrogen and van der Waals bonds [10]. Hemicellulose is a heteropolymer composed of several different types of monosaccharides linked by (1,4)-glycosidic bonds, such as xylose, arabinose, mannose and galactose [11]. The structure of hemicellulose varies greatly among different plants. For example, xylan is the main component of hemicellulose in the cell wall of herbaceous and broad-leaved trees, while mannose is the main component of hemicellulose in cork and conifer wood [12,13]. The various components also lead to the diversity of hydrolases for hemicellulose degradation. Lignin is a rigid aromatic heterogeneous polymer located in the plant cell wall, which is largely composed of phenolic monomers. It mainly provides structural support and forms a natural, impermeable barrier to resist microbial attack and oxidative stress on the plant [14]. Additionally, lignin links with cellulose and hemicellulose by hydrogen and covalent bonds, forming tough and tight biopolymers [15].

Although the abundant lignocellulosic resources are considered as the potential resources for biochemicals and biofuels production, the complex and heterogeneous structure limit the development of lignocellulosic biorefinery and largely decrease the final efficiency of high value-added chemicals and biofuels production [16,17]. Commonly, the lignocellulosic materials should be pretreated first to break its rigid structure and enhance microbial and enzymatic catalysis efficiency. Lignocellulose pretreatment is mainly composed of physical and chemical methods [18,19,20]. The former method mainly reduces the particle size of biomass and increases the specific surface area, which contributes to better mass and heat transfer without the destruction of its interior structure [21]. Recently, irradiation approaches such as microwave and ultrasound techniques are widely used as physical pretreatment of lignocellulose, which can disrupt chemical bonds of the biomass in a short duration under high energy radiations, causing more carbohydrates to be digested [22]. However, these physical processes usually need huge energy consumption, and the effectiveness for biomass structure disruption is still limited [23]. Chemical methods can remove lignin by acid or alkali to reduce the degree of cellulose polymerization to obtain the fermentable sugars [24,25]. Although chemical methods were extraordinarily effective at destruction, the serious environmental pollution and high cost on detoxification and desalination still hinder their further industrial applications [26,27]. Physicochemical pretreatment, such as steam explosion pretreatment, not only can effectively disrupt the crystal structure, but it can also increase the efficiency of enzymatic hydrolysis and sugar yield, which is considered as the most cost-effective pretreatment method [28]. Additionally, supercritical pretreatment is also considered as an economical and environmentally friendly process to replace the conventional pretreatment process. Daza Serna et al. reported that production cost of hydrolysate by acid pretreatment was 15.32 USD/kg, while the cost by supercritical pretreatment was only 1.56 USD/kg [29].

The pretreatment process just breaks the rigid structure of lignocellulose, and further lignocellulose bioconversion still depends on hydrolases and microorganisms. Thus, this review comprehensively summarizes hydrolases and microorganisms for lignocellulosic degradation. In addition, the current situation, bottleneck and future development prospect of biotechnologies to achieve the degradation and conversion of lignocellulose is also introduced, including separated hydrolysis and fermentation (SHF), simultaneous saccharification and fermentation (SSF) and consolidated bioprocessing (CBP), contributing to the further development of the lignocellulosic biorefinery.

## 2. Typical Hydrolases and Microorganisms for Lignocellulose Degradation

### 2.1. Hydrolases Responsible for Lignocellulose Degradation

Hydrolases have been widely used to degrade lignocellulose owing to their ecofriendly and efficient properties, and the global cellulase/hemicellulase enzyme market has been growing sustainably in recent years [30]. These hydrolases can convert complex carbohydrate polymers into available sugars, which is considered as the rate-limiting step in lignocellulosic biorefinery [31]. Until now, the analysis of degradation mechanisms provides the basics for further research on enhancing the degradation efficiency of lignocellulose.

Cellulase has been widely used in many industrial fields, such as food processing, medical materials, detergent processing, etc. [32,33,34]. Generally, endoglucanase (EG), exoglucanase (CBH) and *β*-glucosidase (BG) are the major three components for cellulose degradation [35]. EG acts on the amorphous region of cellulose molecules to degrade *β*-1,4 glycosidic bonds, which will then be hydrolyzed into oligosaccharides. CBH randomly cleaves *β*-1,4 glycosidic bonds on the terminus of cellulose macromolecules, releasing cello-oligosaccharide including cellobiose, cellotriose, cellotetrose, etc. BG hydrolyzes these cello-oligosaccharide cut down by EG or CBH into glucose, which is commonly considered as the rate-limiting factor in the whole cellulose degradation process [36].

Unlike cellulose, hemicellulose possesses more complex structure. Taking xylan as an example, it not only contains xylose, its side chains also contain different kinds of polysaccharides [37,38]. Therefore, the kinds of hemicellulase are complex, which are mainly composed of glycosidase hydrolases (xylanase, xylosidase, arabinosidase, galactosidase) and carbohydrate esterases (acetyl xylan esterase, feruloyl esterase, etc.). Among these enzymes, *β*-xylanase and *β*-mannanase can randomly cleave the interior of hemicellulose backbone structure and produce oligosaccharides. Other glycosidase hydrolases act on side chains to assist hemicellulose hydrolysis and produce monosaccharides or disaccharides [39]. For example, *α*-l-arabinofuranosidases and *α*-l-arabinases co-hydrolyze arabinan side chains into arabinose. Among hemicellulolytic esterases, acetyl xylan esterase acts on acetyl from xylose residues to eliminate hindrance of acetyl group to endoxylanase [40]. Feruloyl esterase can catalyze the hydrolysis of ester bonds between polysaccharide and phenolic acid, promoting the degradation of the cell wall [41].

Lignin degradation is a tough issue for the total component utilization of lignocellulose. Although some chemical pretreatments can effectively remove lignin, the increased cost and produced inhibitors still affect the further applications [42]. Microorganism with the capability of lignin degradation is considered as the appropriate potential chasses to realize the full component utilization of lignocellulose. Ligninase are mainly composed of lignin peroxidases (LiPs), manganese-dependent peroxidases (MnPs) and laccase [43]. These enzymes will dissolve the wax on the plant surface first, and then they will make mycelia enter the plant to secrete various enzymes, triggering a series of free radical chain reactions with the participation of molecular oxygen to make lignin oxidation radically [44]. These enzymes can break lignin structure and decrease the crystalline of cellulose [45]. However, the slower catalytic reaction rate still limits the total efficient utilization of lignocellulosic biomasses, and the mechanisms of lignin degradation need further studies in the future.

Recently, lytic polysaccharide monooxygenases (LPMOs) have been found to deconstruct crystalline polysaccharide and boost the lignocellulosic degradation. It has been proved that LPMOs exhibit good synergistic action on cellulase, which conduce the degradation of small crystalline fibrils [46]. It is worth noting that LPMOs require exogenous electrons to reduce Cu^2+^ to Cu^+^, which will react with O_2_ to converge the copper–superoxide complex to decompose polysaccharide substrates [47]. These exogenous electrons come from reductant, cellobiose dehydrogenase or photocatalyst. The effect on LPMOs activity by electron donor depends on their reduction potential. Frommhagen et al. have studied the effect of 34 reducing agents on the LPMOs activity derived from flavonoids and lignin-building blocks [48]. It has been found that polyphenols such as 1,2-benzenediol or 1,2,3-benzenetriol can promote the LPMOs activity due to their lower redox potential.

In-depth understanding of hydrolase degradation mechanisms will guide the optimization of hydrolase degradation conditions so as to improve the saccharification efficiency [49]. Saccharification efficiency is significantly affected by enzyme loading, pH, temperature, carbon source concentration, etc. Generally, coordinating enzyme load and degradation efficiency are the effective approaches to overcome economic bottleneck for the commercial processing [50]. Xu et al. [51] adapted a fed-batch strategy to optimize enzymatic hydrolysis efficiency. Under the conditions of 22% (*w*/*v*) substrate content and cellulase dosage of only 4 FPU/g dry biomass, the achieved glucose titer and yield were reached 122 g/L and 80%, respectively. Similarly, Gao et al. [52] optimized different proportions of hydrolase, achieving high glucose (around 80%) and xylose (around 70%) yields.

### 2.2. Typical Microorganisms for Lignocellulose Degradation

In nature, approximate 200 species of microorganisms with the capability of lignocellulose degradation have been found in the past 50 years from agricultural waste, ruminant stomach and some of insects. They usually exist in form of microbial consortia to degrade lignocellulose and resist the invasion of external environment. For example, rumen microbial consortium is composed of bacteria, protozoa, fungi, archaea and a small proportion of phages, which forms a complex symbiotic system to participate in the lignocellulose degradation [53]. These rumen microorganisms could anaerobically digest rice straw, and the degradation efficiency of cellulose and hemicellulose reached 46.2% and 60.4%, respectively [54]. Another typical example is termite guts, which possess abundant enzymes to degrade lignocellulosic materials. [55]. Nevertheless, complex synergistic mechanism of lignocellulose conversion between the microbial consortium of core enzymes and strains are not clearly elaborated, which limited the stability and efficiency of lignocellulose degradation.

Compared to microbial consortia, single microorganisms for lignocellulosic biorefinery are more extensively studied due to their relatively clear metabolism and action mechanisms [56,57]. Bacteria have potential applications in the lignocellulose degradation due to its fast growth and tolerance of acid and alkaline conditions [58]. Most of them belong to anaerobic bacteria, such as *Clostridium* spp., *Ruminococcus* spp., *Pseudomonas* spp., *Bacillus* spp., *Proteus* spp. and *Serratia* spp. [59,60,61]. They usually produce enzyme systems for lignocellulose degradation, which comprises cellulase, xylanase and cellobiase. In bacteria, cellulolytic enzymes are organized into complexes, which are called cellulosomes consisting of multiple enzymes to degrade plant cell wall into soluble polysaccharides [62]. They also secrete scaffold protein, an integral cellulose-binding module to bind cellulose. The close substrate contact facilitates the degradation of lignocellulose [63]. *C. thermocellum* is widely used for lignocellulosic conversion due to its abundant degradation enzyme system. It usually grows at a high temperature (50–60 °C), leading to better degradation efficiency compared to medium-temperature bacteria. Additionally, its enzymes can tolerate extreme environments, including phenolic compounds from the pretreated lignocellulose, which can be directly used for lignocellulose degradation without the movement of phenolic compounds [64].

Filamentous fungi can secrete a greater variety of hydrolases compared with bacteria, and the higher enzymatic activities are more suitable for industrial manufacture [65]. In fact, fungi are the major source of cellulase industrial producers, such as *Trichoderma* spp. and *Aspergillus* spp. [66,67]. Among these fungi, *T. reesei* is considered as the king degrader owing to its excellent capability of cellulose degradation, which has become the first choice for the commercial production of cellulase [68]. However, the lack of capacity for *β*-glucosidase secretion leads to the excess accumulation of cellobiose, which will result in the substrate inhibition for further cellulose degradation [69]. Inspired by natural microbial consortia, various complementary co-culturing systems were constructed combining *T*. *reesei* and other fungi possessing excellent *β*-glucosidase secretion capability. For example, *T. reesei* can be co-cultivated with *A. phoenicis* in multispecies biofilm membrane reactors to enhance *β*-glucosidase activity. Compared with the single *T. reesei* biofilm, the microbial co-culture gave a 2.5-fold increase in *β*-glucosidase production [70].

## 3. The Approaches for Bioconversion of Lignocellulose

As described above, various enzymes and microorganisms can degrade lignocellulose into monosaccharides, while the efficient conversion of lignocellulose into desired value-added products is still the final target [71]. Until now, the major methods of lignocellulosic biorefinery include separated hydrolysis and fermentation, simultaneous saccharification and fermentation and consolidated bioprocessing (Figure 2). Advantages and drawbacks of main lignocellulosic biorefinery strategies are listed in Table 1.

### 3.1. Separated Hydrolysis and Fermentation

Separated hydrolysis and fermentation (SHF) can be divided into two steps: preparation of fermentable sugars and conversion of these sugars into biochemicals and biofuels [42]. To obtain pentose and hexose, various hydrolases should be added first. Then, the sugar liquid is separated and transferred to the fermentation system. Lastly, the obtained fermentable sugar is fractionated, derived and re-treated for high value-added product production [72]. For example, Abengoa Bioenergy, the largest commercial cellulosic biorefinery in the world, can produce up to 25 million gallons of ethanol every year. It uses an acid catalyzed steam explosion approach to break the structure of lignocellulose. Then, the cellulose and hemicellulose are degraded into hexoses and pentoses, which can be further fermented into ethanol by SHF. The residual lignin component can provide power to the factory, allowing it to operate as a self-sufficient renewable energy producer [73]. Additionally, Clariant is expected to produce 1000 tons of ethanol per year from residues such as wheat and corn stover, realizing the pretreatment to cellulose saccharification and fermentation from straw [74]. Taken together, owing to the separation of lignocellulose degradation and bioconversion, each step of SHF can be carried out in the optimum conditions, resulting in a high hydrolysis rate and yield, making it easily to scale up production [75,76]. The cost of extra cellulase to degrade per kg lignocellulose is basically more than 0.31 USD. If the price is more competitive with fossil fuel conversion, the cost of bioproduction such as lactic acid converted from lignocellulose should no more than 0.80 USD/kg [77]. This means that the cost of hydrolases still limits the commercial application for lignocellulosic biorefinery. To reduce the cost of enzymes, Wang et al. [78] cultivated *T. viride* and utilized the secreted crude cellulase to degrade cellulose. The final titer of fermentable sugars by crude cellulase achieved 17.32 g/L, reaching 82.2% saccharification efficiency of commercial cellulase. The butanol production by *C. acetobutylicum* with crude cellulase and commercial cellulase were 5.05 g/L and 5.56 g/L, respectively. Compared with the extra addition of commercial cellulase, the crude enzyme did not need separation and purification, which can largely decrease the cost.

The recycle and reuse of crude enzymes is a bottleneck, which results in the waste of fermentation time and enzyme resources. To overcome this limitation, a magnetic nanoparticle has been developed [79]. The surface of iron oxide magnetic nanoparticles was bifunctionalized with silica and amine groups, which can immobilize cellulase, xylanase and *β*-1,3-glucanase onto the iron oxide magnetic nanoparticles, realizing the recycled usage of enzymes. They still maintain activities after six rounds of utilization compared to free enzymes. The approach of enzyme immobilization can remarkably reduce the cost of process, but the particle aggregation decreased the rate of substrate hydrolysis. In addition to the immobilization of hydrolytic enzymes, immobilized strains have also been studied. For example, Zheng et al. [80] immobilized *C. tyrobutyricum* in the macroporous Ca-alginate-lignin beads to produce butyric acid from pretreated corncob. The butyric acid production was basically maintained after 10 repeated batches of fermentation, which decreased the fermentation cost and time of downstream strains. Moreover, compared to the suspension culture, immobilized cells can assume higher stress factor resistance, such as pH and toxic by-products and increase cell growth rate.

Although enzyme and microorganism immobilization can significantly reduce costs and enhance tolerance under high substrate concentrations, the costs associated with the recycling technology and operation in biorefineries are rarely addressed in many studies [81]. On the one hand, the relevant parameters will change with the scale expansion of the biorefinery process, such as the effects on microbial and enzymatic activities caused by the toxic by-products. On the other hand, the cost and efficiency of immobilized enzymes need further economic evaluations [82].

### 3.2. Simultaneous Saccharification and Fermentation

Though SHF is a relatively mature process and has been used to produce many chemicals and biofuels, the separated steps of degradation and conversion increase the equipment costs, prolonged fermentation duration and substrate inhibition. To further optimize this technology, Gauss et al. put forward an idea, in which hydrolysis and fermentation steps were combined in one reactor, called simultaneous saccharification and fermentation (SSF) [83]. This process can relieve the end-product inhibition and enhance the hydrolysis efficiency. Many studies focused on the improvement of SSF productivity by increasing the substrate loads [83]. Nevertheless, high feedstock loading will make the medium thick, obstructing the mass transfer [84]. The enzymatic activity and microbial fermentation will be also restricted by the high feedstock loads. The supplementation of surfactant or soluble polysaccharides can increase the cell permeability and improve the tolerance of microorganisms in high substrate environment. For example, Xiao et al. reported that the addition of polyoxyethylene (80) sorbitan monooleate in SSF could speed the cell growth rate and promote the saccharification efficiency. The fermentable sugars production was increased by 13.5% compared to the control [85].

However, the optimum growth temperature of most microorganisms to produce biochemicals and biofuels is mesophilic, while the ideal temperature for hydrolysis lignocellulose is over 50 °C, which is only suitable for thermophilic microorganism fermentation [86]. *Bacillus subtilis*, a thermotolerant strain, could simultaneously utilize xylose and glucose to produce acetoin. The highest acetoin production reached 12.55 g/L, which was 15.27% higher than that of SHF [87], while the production by moderate and cryogenic microorganisms in the SSF is much lower than that in SHF. Li et al. used *S. cerevisiae* to produce glucaric acid from lignocellulose by SSF and SHF [88]. The glucaric acid produced by SSF was much lower than SHF. The optimal fermentation temperature of *S. cerevisiae* (30 °C) limited the rate of hydrolysis. Moreover, the disinfection of by-products produced in the pretreatment step will lead to higher concentrations of lignin, which will nonspecifically bind to cellulase and lower the cellulase activity. To solve this obstacle, genetic engineering and mutation can be carried out to obtain strains with high temperature tolerance; thus, the hydrolysis and fermentation temperature can be harmonized. Recently, Wu et al. [89] metabolically constructed *C. acetobutylicum* L7 by homologous overexpression of *glcG*, which can tolerate high temperatures A total of 10.8 g/L of butanol can be produced from 48 g/L corn stover at 42 °C, increased by 40% compared with the original strain.

### 3.3. Consolidated Bioprocessing

As discussed above, both SHF and SSF should add extra hydrolases, which will limit their industrial application. Consolidated bioprocessing (CBP) is a multi-step process in one bioreactor, including hydrolase production, enzymatic hydrolysis and microbial fermentation. The process can eliminate the exogenous enzyme addition, which can reduce the complexity and costs for efficient lignocellulose conversion. Compared with SHF and SSF, CBP can reduce the cost of lignocellulosic biotransformation by about 78% [90]. As the lignocellulosic hydrolase systems and product metabolic pathway are both complex, the increased metabolic loads will obviously hinder the final efficiency of production. Thus, except the single bacterium fermentation, the co-culture strategy via CBP is attracting more and more attention.

#### 3.3.1. Single Microorganism Strategy

The CBP strategy directly completing hydrolysis and conversion of lignocellulose in one single microorganism can be divided into two approaches, including the native and recombinant strategy (Figure 3) [91]. The former is expressing the metabolic pathway of target products through genetic engineering in the native cellulolytic microorganism. The major challenge in the native strategy is to maintain the high hydrolytic ability and improve the productivity of the desired product at the same time. To enhance the final butanol production from cellulose via CBP in *C. cellulovorans*, a metabolic engineering approach based on a push–pull strategy was developed by Wen et al. [92]. The trans-enoyl-coenzyme A reductase was overexpressed to pull carbon flux from acetyl-CoA to butyryl-CoA. Then, an acid reassimilation pathway uncoupled with acetone production was introduced to redirect the carbon flow from butyrate and acetate towards butyryl-CoA. Through this engineering, the final butanol production was increased by 135 folds compared with that of the wild type. *C. thermocellum* is another cellulose degrader without the butanol production capability. Through the introduction of exogenous key enzymes related with butanol production into *C. thermocellum*, the engineered strain was able to produce 357 mg/L of n-butanol from cellulose within 120 h [93]. The recombinant strategy is introducing cellulase or/and hemicellulase into the non-cellulolytic strains, which can confer them with the ability for lignocellulosic degradation [94]. Compared to the native strategy, these strains have good capacity of target product conversion. The major challenge is how to enhance the capability for lignocellulosic degradation. Chen et al. [95] constructed a two cell-surface displayed yeast for cellulosic ethanol conversion, which can heterologously express cellulases and xylanases. After the metabolic engineering, the *S. cerevisiae* consortium produced 1.61 g/L of ethanol from 20 g/L steam-exploded corn stover without the exogenous hydrolase addition.

However, a common bottleneck of these two strategies in single microbial fermentation is the lower product productivity. The major reason is that the simultaneous degradation and conversion of lignocellulose in one single microorganism will lead to severe metabolic burdens. For instance, the metabolic engineering of cells will expend intracellular energy to generate coenzyme factor, which leads to less energy for cell growth [96]. Usually, removement of unnecessary genes and enhancement of target product metabolic pathway will improve the capacity for target product production and decrease by-products. In contrast, unreasonable engineering modification will cause “intermediate toxicity” or “low enzyme activity”, upsetting normal cellular processes [97]. *Corynebacterium glutamicum* is a well-known microbe to produce succinic acid. To enhance its capacity for succinic acid production, *pntAB* was integrated into *C. glutamicum* to increase the NADH secretion. Additionally, it has been investigated that *pgi* from *C. glutamicum* will make carbon flux flow into the Embden–Meyerhof pathway, leading to the increase in by-products. To decrease the by-product production, *pgi* was deleted to redirect carbon flux in pentose phosphate pathway. However, it has been found that *pgi*-deficient *C. glutamicum* grew poorly with glucose as the sole substrate, which was caused by the downregulation of *ptsG* in *pgi*-deficient *C. glutamicum* responsible for the transport of glucose molecules. To solve this problem, *ptsG* was integrated into *pgi*-deficient strain. Finally, compared to control strain with the yield of 1.07 mol/mol glucose, the succinate yield of engineered strain reached 1.37 mol/mol glucose [98]. Even so, the improvement of productivity is limited. Furthermore, many microorganisms still lack the complete knowledge of metabolism machinery and mature molecular tools for genomic editing, which limits their applications to meet commercial demands [97]. Additionally, the complex of hydrolyses systems and metabolic pathways will increase metabolic burdens for single microorganisms, and relatively low production and yield are still obtained even in the engineering strains via CBP.

#### 3.3.2. Microbial Co-Culturing Systems Construction

In nature, organisms have difficulties in performing a large number of tasks alone in the intricate environment. They must pull together and perform their duties for survival, which exist as various of biological organizations, such as excrement of animals, sludge, rivers and so on [99]. Inspired by this, lignocellulose hydrolysis and conversion can also be divided into different strains, which will alleviate cell metabolic burdens through functional specialization [100]. However, their metabolic end-products are usually CH_4_, H_2_ and a little organic acid [101]. Therefore, how to establish synthetic microbial communities applying to industrial manufacture is a critical research topic.

Ambiguity tasks in microbial consortium will create competition for nutrients, resulting in the failure of a stable fermentation system [102]. Thus, the specific labor division between cells in the co-culturing systems is necessary for the efficient lignocellulose degradation and conversion. For lignocellulosic biorefinery, the feasible consolidated bioprocess is that lignocellulose is degraded by upstream strain, followed by fermentable sugars conversion by downstream strain. Especially, the rapid consumption of fermentable sugars can relieve the substrate inhibition to improve upstream strains hydrolysis [103]. For example, a microbial co-culturing system containing lignocellulose degrader *T. thermosaccharolyticum* M5 and butanol producer *C. acetobutylicum* NJ4 successfully achieved butanol production from xylan and unpretreated corncob [104]. The secreted xylanase and xylosidase by strain M5 degrade xylan to xylose, and the rapid consumption of xylose by strain NJ4 relieved the inhibition on xylanase. Furthermore, strain M5 can produce butyrate, which can be further assimilated by strain NJ4 to produce butanol. These two strains can generate well supplementary interaction, making fermentation process more efficiently, and 7.61 g/L of butanol was directly produced from corncob. Similarly, a microbial co-culturing system containing *A. succinogenes* 130Z and *T. thermosaccharolyticum* M5 also achieved succinic acid production from lignocellulosic materials [105]. After single-element and response surface optimization, the optimal production of succinic acid was 32.50 g/L and 12.5g/L from 84 g/L xylan and 80 g/L corncob, respectively. *C. thermocellum* is another thermophilic bacterium with the capability of cellulose degradation. Chi et al. [106] constructed a co-culturing system containing *C. thermocellum* and *C. thermobutyricum* to produce butyric acid. Metabolic analysis indicated that sugar could be released by *C. thermocellum* and converted to butyric acid by *C. thermocellum* rapidly, in which the yield of butyric acid reached 33.9 g/L from rice straw. In addition, the secondary metabolism of *C. thermobutyricum* also lead to the hyper-production of butyric acid, leading the reassimilation of by-products such as acetic acid and ethanol.

Similar growth conditions are necessary for the efficient lignocellulosic biorefinery in the microbial co-culturing system. *T. reesei* and *Ustilago maydis* can both grow in oxygen conditions at 30 °C. In this co-culturing system, *T. reesei* plays a role in lignocellulosic degradation, and *U. maydis* is responsible for itaconic acid production. The itaconic acid titer achieved 33.8 g/L from 120 g/L α-cellulose with fed-batch CBP strategy [107]. To further improve the adaptation between members of the co-culturing system, genetic engineering and adaptive evolution are adopted. For example, Wen et al. [108] designed a co-culture system to produce butanol from lignocellulose by *C. cellulovorans* and *C. beijerinckii*. Butanol fermentation preferred low pH by *C. beijerinckii*, while *C. cellulovorans* cannot grow well at a pH below 6.4. Thus, *C. cellulovorans* was engineered to enhance the tolerance of low pH, which can improve lignocellulosic saccharification and butanol fermentation. The engineered consortium finally produced 3.94 g/L butanol, which was five times higher than the control.

The substrate degradation rate is still the key rate-limiting step in the CBP process. Thus, the acceleration of lignocellulose degradation rate and improvement of fermentable sugars’ releasing rate are necessary for the CBP process. Generally, fungi possess the higher lignocellulose degradation capability than bacteria [109]. However, the contradiction of growth conditions including temperature and oxygen demands between aerobic fungus and anaerobic members is also a challenge for lignocellulosic biorefinery. Thus, multispecies biofilm membrane reactors have been exploited, which can create the aerobic and anaerobic environment at the same time. Firstly, filamentous fungi will easily attach to the surface of the material due to its good film formation property. As time goes on, the cells occupy the entire surface of the material and form a dense biofilm. Then, cells on the biofilm will consume oxygen from the air and prevent oxygen from penetrating into the liquid phase. A typical successful example is the construction of artificial cross-kingdom consortium in a biofilm reactor containing aerobic fungus *T. reesei* and anaerobic bacteria *Lactobacillus pentosus* [110]. *T. reesei* formed biofilm on the support material to consume oxygen, and the secreted cellulase can efficiently degrade cellulose. Anaerobic bacteria *L. pentosus* grew in the obtained fermentable sugars to produce lactic acid. As a result, the lactic production achieved 19.8 g/L from non-detoxified steam-pretreated beech wood. Furthermore, the complex, stable and efficient lactate producing co-culturing system can be used as a platform to further produce short-chain fatty acids [111].

## 4. Conclusions and Prospects

Currently, the lignocellulose pretreatment and bioconversion strategy have achieved much advancement. The high cost and low efficiency of the lignocellulosic biorefinery still limit its further industrial application. In Europe, there were 224 biorefineries operating in 2017. While most of them produced biofuels from food grade feedstocks, only 43 factories used lignocellulosic feedstocks [112]. Up to now, several effective biotechnologies have been applied in large-scale production to achieve lignocellulosic biorefinery, such as SHF and SSF. However, the toxic by-products generated by pretreatment processes and high costs of various hydrolases still limit further development. Compared with SHF and SSF, CBP is considered as a promising approach for lignocellulosic biorefinery without the extra addition of various hydrolyses. However, due to the heavy metabolic burdens in single microorganisms, lower efficiency also limits its commercial application. Inspired from natural microbial consortia, microbial co-culturing systems can divide lignocellulose degradation and conversion into different microorganisms through labor division, becoming a potential economic way to achieve efficient lignocellulosic biorefinery. If this technology develops maturity and successfully explores a commercial road, it will fundamentally free people from their dependence on fossil resources and change lifestyles.

## Figures and Tables

**Figure 1 molecules-26-05411-f001:**
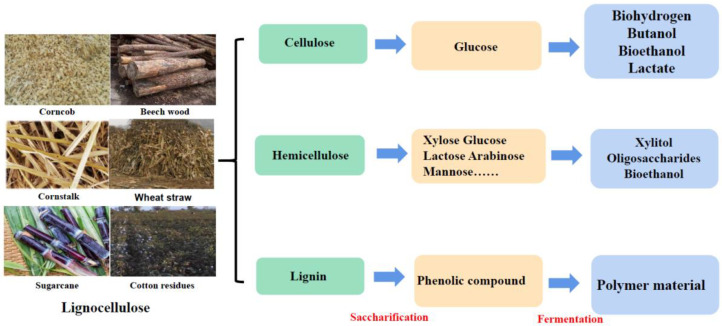
Types and structure of lignocellulose and the value-added products converted by lignocellulose.

**Figure 2 molecules-26-05411-f002:**
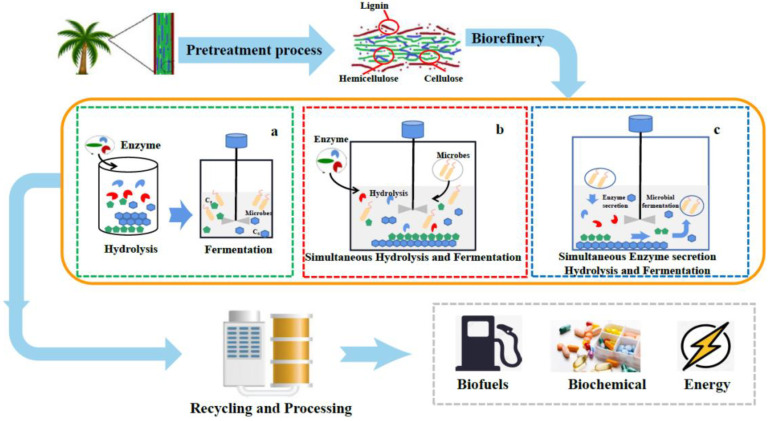
The strategies for lignocellulosic biorefinery. (**a**) separated hydrolysis and fermentation; (**b**) simultaneous saccharification and fermentation; (**c**) consolidated bioprocessing.

**Figure 3 molecules-26-05411-f003:**
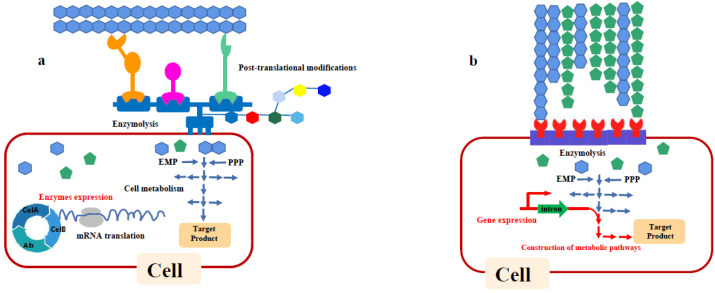
Biorefinery strategy in single microorganism system. (**a**) Recombinant cellulolytic strategies; (**b**) native cellulolytic strategies.

**Table 1 molecules-26-05411-t001:** Advantages and drawbacks of main lignocellulosic biorefinery strategies.

	Process Steps		Advantages	Drawbacks
SHF	Adding exogenous hydrolase Enzyme hydrolysis Sugar fermentation		Both saccharification and fermentation can be carried out under the best reaction conditions of pH and temperature	Divided into two steps to increase the process complexity and equipment cost High enzyme cost Glucose accumulation leads to end product inhibition
SSF	Adding exogenous hydrolase Simultaneous enzyme hydrolysis and sugar fermentation		No hydrolase inhibition Reduce unnecessary equipment Simplified operation steps	Saccharification and fermentation cannot be carried out under the best reaction conditions
CBP	Single microorganism strategy	Simultaneous production of hydrolase, enzyme hydrolysis and sugar fermentation	No hydrolase inhibition No additional enzyme costs	High metabolic burden Need complex molecular modification
	Microbial co-culturing systems		No hydrolase inhibition No additional enzyme costs Low metabolic burden	Conditions of microbial consortium is difficult to coordinate

## Data Availability

Data is contained within the article.

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
