# Peer review of "Challenges and Future Perspectives of Promising Biotechnologies for Lignocellulosic Biorefinery"

_molecules, 2021, doi:10.3390/molecules26175411_

Round 1

Reviewer 1 Report

The current title of the article is too broad and suggests a comprehensive approach to the subject of biorefineries. In the new title, please underline that the review focuses on biological methods

Lack subsection clearly identifies future challenges for this type of biorefineries. The authors limited themselves to a short summary.

Lack no information on the cost of solutions and comparisons of the strengths and weaknesses of the different methods

“The former method mainly reduces the particle size of biomass and increases the specific surface area, which contributes to better mass and heat transfer without destruction of its interior structure1” – editing error please correct no. 1 (1 – reference?).

Author Response

Response to Reviewer #1:

Reviewer #1:

  1. The current title of the article is too broad and suggests a comprehensive approach to the subject of biorefineries. In the new title, please underline that the review focuses on biological methods

In this manuscript, the title of the article has been revised to “Challenges and future perspectives of the promising biotechnologies for lignocellulosic biorefinery”, in which highly underline the biological methods.

  1. Lack subsection clearly identifies future challenges for this type of biorefineries. The authors limited themselves to a short summary.

The challenges and prospective development direction of lignocellulosic biorefinery have been introduced in the part of Conclusion and prospect in Lines 554-572.

  1. “The former method mainly reduces the particle size of biomass and increases the specific surface area, which contributes to better mass and heat transfer without destruction of its interior structure1” editing error please correct no. 1 (1 reference?).

The error has been corrected.

  1. Lack no information on the cost of solutions and comparisons of the strengths and weaknesses of the different methods

Among these strategies, SHF divides hydrolysis and fermentation into two individual steps. Both of them could be carried out in the optimum conditions, contributing to the high rate and yield hydrolysis and bioconversion. However, the expensive enzyme and substrate inhibition limit its development. SSF integrates enzymatic hydrolysis and fermentation in one step, which alleviates hydrolase inhibition and save equipment cost. Nevertheless, the ideal temperature for hydrolysis lignocellulose is over 50 ℃, which is only suitable for thermophilic microorganism fermentation. CBP integrates enzyme production, enzymatic hydrolysis and fermentation in one step. It is considered as the promising economic approach for lignocellulosic biorefinery owing to it does not need extra addition of expensive hydrolases. While single microorganism system will bear heavy metabolic burdens limiting the final efficiency. The co-culturing systems can assign heavy metabolic work into different microorganisms through labor division, becoming a potential approach for lignocellulosic biorefinery. The coordination for different growth conditions and complex relationship in consortium need more researches. Thus, the efficiency of co-culturing systems via CBP is commonly lower than that by SSF or SHF. The comparisons of the strengths and weaknesses of the different methods have been described and added in Lines 311-320, 345-353, 362-364, 376-379, 407-411, 444-450 and 483-488. Additionally, in order to better understand for authors, an extra Table1 has also been concluded. Furthermore, the costs of solutions have been described and added in Lines 121-123 and 315-318, including the costs of pretreatment processes and extra addition enzymes.

Reviewer 2 Report

I my opinion the work entitled “Challenges and future perspectives in lignocellulosic biorefinery” gives interesting information about lignocellulose degradation and fermentation for biochemicals and biofuels production, therefore I think it is possible to publish in this journal. The work is very well written but there are words with different typography, check this please.

Author Response

Response to Reviewer #2:

Reviewer #2:

  1. In my opinion the work entitled “Challenges and future perspectives in lignocellulosic biorefinery” gives interesting information about lignocellulose degradation and fermentation for biochemicals and biofuels production, therefore I think it is possible to publish in this journal. The work is very well written but there are words with different typography, check this please.

Thanks for your opinion, and the words typography has been carefully checked and unified.

Reviewer 3 Report

Review of molecules-1320778 Challenges and future perspectives in lignocellulosic biorefinery

This review article discusses various pretreatment approaches for lignocellulosic biomass, and various bioprocesses for the conversion of lignocellulosic biomass to biofuels or value-added products. The manuscript fails to inform the reader regarding the current status of various technologies for the development of commercial processes, and the authors appear to be unaware of what product yields are required for a successful commercial process. The lack of line numbers for the text makes it cumbersome to provide specific feedback.

There are numerous review articles concerning the pretreatment and/or microbial conversion of lignocellulosic biomass and this manuscript fails to discuss what limitations exist in that prior literature that justifies the need for an additional review. There have been several efforts to develop commercial processes for the utilization of lignocellulosic biomass. Some of those efforts did not succeed while other efforts have resulted in commercial processes operating today. The current manuscript ignores these facts and the lessons these efforts afford to those seeking to develop improved processes. This oversight renders this review article irrelevant. Rejection of this manuscript is recommended.

Author Response

Response to Reviewer #3:

Reviewer #3:

This review article discusses various pretreatment approaches for lignocellulosic biomass, and various bioprocesses for the conversion of lignocellulosic biomass to biofuels or value-added products. The manuscript fails to inform the reader regarding the current status of various technologies for the development of commercial processes, and the authors appear to be unaware of what product yields are required for a successful commercial process. The lack of line numbers for the text makes it cumbersome to provide specific feedback.

Until now, the major biotechnologies for lignocellulosic biorefinery includes separated hydrolysis and fermentation (SHF), simultaneous saccharification and fermentation (SSF) and consolidated bioprocessing (CBP). In the SHF process, the process of the enzymatic hydrolysis and fermentable sugars bioconversion are divided into two individual steps. Thus, both of them could be carried out in the optimum conditions, contributing to the high rate and yield hydrolysis and bioconversion. Additionally, this process does not need complex metabolic analysis and engineering, such as metabolic pathway analysis, genetic engineering modification, enzyme expression, etc. Hence, most of biorefineries adopt this approach, but the relatively highly cost of enzyme and the increase cost and complicated operation of separated facility both limit the development of its industrialization. Even SSF process can address long fermentation time and high facility cost in SHF, the mismatching of optimal conditions on hydrolases for lignocellulose degradation and fermentation for sugars bioconversion largely decrease the efficiency of lignocellulosic biorefinery. In addition, the highly cost of various hydrolases is another limited factor for its industrial application. CBP is considered as the promising economic approach for lignocellulosic biorefinery, in which combines hydrolase production, enzymatic hydrolysis and microbial fermentation in one pot. However, the heavy metabolic burdens in single microorganism limit the final efficiency. Thus, it still needs more time and researches to march toward commercialization. Current status of various technologies was added in Lines 311-320, 345-353, 362-364, 376-379, 407-411, 444-450 and 483-488.

There are numerous review articles concerning the pretreatment and/or microbial conversion of lignocellulosic biomass and this manuscript fails to discuss what limitations exist in that prior literature that justifies the need for an additional review.

In order to better discuss the limitations in lignocellulosic biorefinery, more research articles have been added in the manuscript. This manuscript focuses more attentions on discussion of biotechnologies instead of commercial processes in lignocellulosic biorefinery. For lignocellulosic biorefinery, the efficiency of degradation and conversion both affect the final biochemicals and biofuels production. Thus, the discussion and comprehensive summarization of lignocellulosic biorefinery in this manuscript are mainly divided into two parts. The first part focuses on the hydrolases and microorganisms for lignocellulose degradation. The next part of “approaches for bioconversion of lignocellulosic biomass” put more emphasis on the bioconversion of sugars. It is worth pointing out that consolidated bioprocessing (CBP) is considered as the promising approach for lignocellulosic biorefinery, in which combines hydrolase production, enzymatic hydrolysis and microbial fermentation in one pot. However, due to the heavy metabolic burdens, the single microorganism is difficult to simultaneously exhibit well capability of lignocellulose degradation and sugar bioconversion. Co-culturing systems have concerned more and more attention in recent years owing to it can assign heavy metabolic work into different microorganisms through labor division. Hence, the co-culturing systems via CBP may become a potential approach for lignocellulosic biorefinery in the future. Additionally, this strategy still can not meet the requirement for commercial applications, and the related discussion has been added in Lines 479-552.

There have been several efforts to develop commercial processes for the utilization of lignocellulosic biomass. Some of those efforts did not succeed while other efforts have resulted in commercial processes operating today. The current manuscript ignores these facts and the lessons these efforts afford to those seeking to develop improved processes. This oversight renders this review article irrelevant. Rejection of this manuscript is recommended.

The first generation biorefinery commonly uses sugar-based agricultural materials as the feedstocks for biofuels and biochemicals production. This is a relative mature commercial process. Currently, there are almost 370 plants focusing on the first generation biorefinery in Brazil with a total capacity of 43,105 million liters ethanol from sugarcane. In EU, the biodiesel production achieved about 11 million tonnes from various vegetable oil per year. While the large demands for agricultural materials will lead the unstable production costs, the second biorefinery has concerned more and more attentions. Some demonstration plants have been put into operation. For example, Abengoa Bioenergy, the largest commercial cellulosic biorefinery in the world, can produce up to 25 million gallons ethanol per year. Clariant, a German biorefinery company, can produce 1,000 tons of ethanol from lignocellulosic wastes per year. However, these biorefineries still face with some challenges such as enzyme cost, substrate inhibition and low product yield etc., leading biorefineries difficult in further expanding production scale. These descriptions have been added in Lines 51-63 and 296-320. As described in Q2, this manuscript focuses more attentions on biotechnologies for lignocellulosic biorefinery, including their advantages and disadvantages. In addition, the challenges of commercial applications for lignocellulosic biorefinery also discussed in the part of Conclusion and prospect, which have been added in Lines 554-575.

Round 2

Reviewer 3 Report

It is unclear what contribution to the existing literature is made.